# Apple Flour in a Sweet Gluten-Free Bread Formulation: Impact on Nutritional Value, Glycemic Index, Structure and Sensory Profile

**DOI:** 10.3390/foods11203172

**Published:** 2022-10-11

**Authors:** Rita Beltrão Martins, Maria Cristiana Nunes, Irene Gouvinhas, Luís Miguel Mendes Ferreira, José Alcides Peres, Ana Isabel Ramos Novo Amorim Barros, Anabela Raymundo

**Affiliations:** 1CITAB—Centre for the Research and Technology of Agro-Environmental and Biological Sciences, Universidade de Trás-os-Montes e Alto Douro, 5000-801 Vila Real, Portugal; 2CQVR—Centro de Química de Vila Real, Universidade de Trás-os-Montes e Alto Douro, 5000-801 Vila Real, Portugal; 3LEAF—Linking Landscape, Environment, Agriculture and Food, Instituto Superior de Agronomia, Universidade de Lisboa, Tapada da Ajuda, 1349-017 Lisbon, Portugal

**Keywords:** gluten-free sweet bread, apple flour, dough rheology, antioxidant capacity, glycemic index, starch hydrolysis

## Abstract

Baking bread without gluten presents many challenges generally related with poor sensorial and nutritional characteristics, and strategies to overcome this issue are needed. Despite many gluten-free (GF) bread studies, to the best of our knowledge, few are dedicated to sweet GF bread. Sweet breads have traditionally been an important type of food and are still frequently consumed worldwide. Apple flour is naturally GF, and is obtained from apples which do not accomplish market quality requirements and are being wasted. Apple flour was, therefore, characterized in terms of nutritional profile, bioactive compounds, and antioxidant capacity. The aim of this work was to develop a GF bread with incorporation of apple flour, in order to study its effect on nutritional, technological, and sensory characteristics of sweet GF bread. Additionally, in vitro starch hydrolysis and glycemic index (GI) were also analyzed. Results demonstrated the influence of apple flour in dough’s viscoelastic behavior, increasing G’ and G’’. Regarding bread characteristics, apple flour led to better acceptance by the consumer, with firmness increasing (21.01; 26.34; 23.88 N), and consequently specific volume decreasing (1.38; 1.18; 1.13 cm^3^/g). In addition, an increase of bioactive compounds content and antioxidant capacity of the breads were revealed. As expected, the starch hydrolysis index increased, as well as GI. Nevertheless the values were really close to low eGI (56), which is a relevant result for a sweet bread. Apple flour showed good technological and sensory properties as a sustainable and healthy food ingredient for GF bread.

## 1. Introduction

Worldwide, at least 1% of the population lives with Celiac Disease (CD); the prevalence of this disease varies according to the geographical area, and can reach 2.6% [1]. Additionally, about 1% to 15% of the world’s population is affected by other gluten and wheat related disorders, meaning that 2% to 16% of consumers in the world have the need to adopt a strict gluten-free (GF) diet, as the only known solution to these health conditions [2]. Moreover, it is well known that the number of consumers who choose to avoid gluten containing products, for many different reasons, is growing each year, making GF products an important market trend [3,4]. According to Markets and Markets’ gluten-free products analysis (2020), forecasts point to a global market growth from 5.6 billion US dollars in 2020 to 8.3 billion US dollars in 2025, with bakery products representing the highest market share [5].

Gluten plays an essential role in normal breadmaking, as it is responsible for the viscoelastic behaviour of the dough, higher volume, and softer texture of wheat breads [6]. Thus, gluten-free bread (GFB) faces many technological and rheology challenges, influencing its sensory characteristics and consequently consumer acceptance [4,7]. Furthermore, GFB often presents unbalanced nutritional profile, bringing many constraints to a celiac patient’s diet, and consequently to their health [7,8], since celiac patients need to avoid gluten containing food products. This health condition is often associated with type I diabetes mellitus, which leads us to the importance of finding GF sweet bread formulations with low glycemic index (GI) and without adding refined sugar, in order to provide alternatives to consumers who are looking for GF products [2,8]. Nevertheless, despite all the efforts that have been made to develop GF bread formulations, to the best of our knowledge, just few researches have been dedicated to sweet bread. Thus, it is important to focus on GF flours with low GI, in order to be suitable for celiac patients with diabetes, but also to any other consumer, since the World Health Organization (WHO) recommends reducing ingestion of added and free sugars [9].

Sweet breads are a traditionally important type of food, and many different countries around the world have numerous recipes, having as main ingredients: wheat flour, a sweetener—sugar or honey, fat, and spices—cinnamon or fennel [10]—recipes which do not allow celiac patients to consume them. As an important type of bread still, sweet breads are commonly consumed in many places across Europe [10], in Iran [11], and in Mexico [12], among other countries, representing an important bakery product.

The use of by-products has been considered by many authors as a good solution to improve GF bread’s nutritional, functional and technological quality [3,4,13,14]. Many studies have shown promising results for developing GF bakery with the addition of by-products from the fruit industry, such as apple pomace powder [15,16,17,18,19,20], and apple fiber [21,22], since their functional properties represent an important contribution to improvement of GFB quality. Nevertheless, as far as we know, apple flour has been studied in gluten containing pasta [23], cookies [24], cakes [25], and muffins [26] but not yet in GFB. Apple flour is obtained from small caliber apples, that are rejected from the supply chain due to size, skin, or any other quality requirements. These apples can be dried and ground into flour, allowing them to be used in the bakery industry. This is a way of upcycling a by-product, contributing for the circularity of the system and reducing food waste [13,27]. Numerous studies have shown various positive apple health outcomes, attributed to its characteristics such as high level of fiber, polyphenols content, antioxidant activity, glycemic index reduction, and anti-obesity effects [28,29,30,31,32]. Apple flour has the advantage of adding natural sugar from the fruit—natural fructose, making possible to obtain a sweet bread and also contribute to lower the eGI of the bread. Other sweeteners that have been used in bakery, such as stevioside [11], polydextrose, fructo-oligosaccharides, maltodextrin, tagatose, polyols, and non-nutritive sweeteners (aspartame, acesulfame-K, sucralose, among others) present the disadvantages of a much more complex production processes, and some of them also have health disadvantages (reviewed in [33]).

In the present work, the flours selected for the formulation with apple flour were buckwheat flour (BWF), sweet potato flour (SPF), and lupin flour (LF), owing to its good performance in GF bread formulation and also low GI. Southgate et al. [34], and also in our previous research Beltrão Martins et al. [35,36], achieved very promising results with BWF addition to GFB, obtaining health benefits, and improving the nutritional and physical profile of the bread. Regarding SPF, partial replacement of wheat flour in sweet bread, general bakery and confectionary products has revealed its improvement qualities [37]. Franco et al. [38] replaced rice flour by SPF in GFB, obtaining good results with 25% incorporation of SPF, when compared to 100% rice flour. Moreover, SPF has a low GI (<55), which represents an advantage when developing GF bakery [39]. As regards LF, its addition to bread increased the fiber and protein content, and decreased carbohydrates. Similarly, LF has shown the ability to decrease GI which goes towards our objective [40]. Additionally, LF is recommended to be used in GF bakery, due to its functional, nutritional and rheological characteristics, strongly related to the functional properties of its proteins [40].

The aim of the present work was developing a GF common bread and analyzing the effect of apple flour incorporation in the nutritive value, antioxidant capacity, in vitro starch hydrolysis and predicted glycemic index, technological quality, and sensory analysis of the GFB. Moreover, converting the GFB into a sweet bread using a sustainable source of sugar.

## 2. Materials and Methods

### 2.1. Raw Materials and Reagents

Ingredients used for bread formulation were: buckwheat flour (Próvida, Pêro Pinheiro, Portugal), white lupin flour (Próvida, Pêro Pinheiro, Portugal), sweet potato flour (Cem Por Cento, Ignoramus, Samora Correia, Portugal), apple flour (Terrius, Marvão, Portugal), dried yeast (Fermipan^®^, Lesaffre, Marcq-en-Baroeul, France), gelling agent hydroxipropylmethylcellulose—HPMC (Wellence^TM^ 321, DuPont, Wilmington, DE, USA), sugar, sunflower oil, salt and water. All ingredients were purchased from the named companies, except sugar, yeast, salt and oil, which were purchased in the local market.

The reagents used were: maleic acid, and petroleum ether acquired from Scharlau (Scharlab, S.L. Barcelona, Spain). Standards for starch (93%), D-glucose and resistant starch (52.5%), together with GOPOD reagent were purchased from Megazyme (Megazyme, Dublin, Ireland). Folin–Ciocalteau’s reagent, acetic acid—glacial extra pure (99.7%), ethanol, potassium persulfate, sodium hydroxide, methanol, hydrochloric acid, iron (III) chloride, sodium acetate, sodium carbonate, sulphuric acid (98%), and potassium hydroxide (98%) were acquired from Panreac (Panreac Química S.L.U., Barcelona, Spain). α-amylase from porcine pancreas (type VI-B, ≥5 units/mg), 2,2-diphenyl-1-picrylhidrazyl radical (DPPH), 2,2-azino-bis(3-ethylbenzothiazoline-6-sulphonic acid) diammonium salt (ABTS), 2,4,6-Tripyridyl-s-Triazine (TPTZ iron reagent), and calcium chloride dihydrate were obtained from Sigma-Aldrich (Sigma-Aldrich Produktions GmbH, Steinheim, Germany). 3,4,5-trihydroxybenzoic acid (gallic acid), 6-hydroxy-2,5,7,8-tetramethylchroman-2-carboxylic acid (Trolox), and 2,4,6-Tripyridyl-s-Triazine (TPTZ iron reagent), were purchased from Fluka (Fluka Chemika, Neu-Ulm, Switzerland). Finally, amyloglucosidase 1100 BG was acquired from Novozymes (Novozymes, Bagsvaerd, Denmark).

### 2.2. Apple Flour Characterization

Apple flour was prepared by Terrius company without removing skin or seeds, from a Portuguese apple cultivar named “Bravo de Esmolfe”. Apples were sliced, dried (40 °C for 7 h) and ground. Apple flour was sieved to obtain a particle size of approximately 0.16 mm to achieve a uniform particle size. Apple flour (AF) was characterized in terms of nutritional profile, bioactive compounds and antioxidants (using the same methods described in the following sections for analysis of bread). The samples were taken from 3 different packages of 2.5 kg, from 2 different batches. All analyses were performed at least in triplicate.

### 2.3. GF Bread Formulations and Sampling

Based on a GFB formulation from our research group [35], preliminary tests were performed in order to develop a new sweet GFB formulation, using low GI flours, reaching a final promising formulation. The control bread formulation has buckwheat, sweet potato and white lupin flours, with hydroxipropylmethylcellulose (HPMC) as thickening agent. The recipe was developed drawing upon previous testing, and also following several successful results from different studies with GFB and alternative flours, mentioned in the introduction. The percentage of each flour was adjusted in order to balance the taste, texture characteristics and volume of the bread. LF was chosen to increase protein and fiber, but due to its bitter flavor needed to be in a well-adjusted share with the other flours. Since the objective was to obtain a sweet bread, the other flour was sweet potato, replacing rice flour, from the initial formulation, and BWF was still included. Moreover, cinnamon was added, since sweet bread traditional recipes mostly have this ingredient. It is important to mention that the sugar added on the same level for all bread formulations was specifically used to allow the process of activating the yeast, since glucose is taken up first, due to the higher affinity of the transporters for glucose, although glucose and fructose are transported through the same transporter [41]. Thus, fructose from AF will not interfere in the fermentation process.

To study the impact of apple flour, its level of incorporation was 17% and 23% of total flours, replacing buckwheat flour. The apple flour level was selected after developing trials with different levels of incorporation, and analyzing the behavior of the dough and the bread, focusing on what could be maximum amount of apple flour, allowing feasible baking to obtain bread, which was 23%. This incorporation level is considered as a “technological limit” for the developed formulation. The other level was selected after lower levels had been tested, since levels below 17% did not present relevant sensory impact (results not shown). All the formulations are summarized in Table 1. Each bread was prepared with a batch of total flour of 400 g, and baking tin dimensions (in cm) were: 6.2 height; 22.5 length; 9.2 width.

The moisture of each flour was determined through an automatic moisture analyzer PMB 202 (Adam Equipment, Oxford, MS, USA) with the following results: BW—10.62%; SPF—7.22% LF—6.12%; and AF—3.68%. Thermo processor equipment (Bimby–Vorwerk, Wuppertal, Germany) was used to mix ingredients. Firstly, the yeast was activated, with the addition of distilled water, yeast and sugar for 2 min at 27 °C at velocity 1 (gentle agitation). The other ingredients were added and mixed for 10 min with a dough mixing program (wheat ear symbol); the dough was poured in a lubricated aluminum pan, and fermented for 50 min at 29 °C in a fermentation chamber Arianna XLT133 (Unox, Cadoneghe, Italy). Finally, the dough was baked for 50 min at 180 °C in an electric oven, model Johnson A60 (Johnson & Johnson, New Jersey, NJ, USA), and allowed to chill for 2 h, at room temperature.

For each formulation, three breads were prepared, and all analyses were performed at least in triplicate: three measurements were taken from each slice individually, in three independent slices from each bread. Mean values and standard deviation measured from each determination were recorded.

### 2.4. GF Dough Properties

#### 2.4.1. Mixing and Pasting Properties

In order to obtain the mixing properties, flour blends’ water absorption (WA) adjustment was performed in a micro-doughLAB 2800 (Perten Instruments, Sidney, Australia), at 14% moisture basis. Using manufacturer’s farinographic protocol at slow 63 rpm (AACCI Method 54-21.01) [42], an optimum dough consistency was reached for all formulations. The water content was chosen by testing different water hydrations to produce a control bread with good quality, based on bread volume and crumb firmness (preliminary assays not presented). The peak value of torque of the optimized control formulation was used as a reference (48 mN·m). According to each flour blend moisture, and considering a 14% moisture basis, the water absorption adjustment represents the following amount of water (in g 100g^−1^ in relation to flours) 107.5; 95.5; 93.7, respectively, for the formulations C, A17, A23, and also presented in Table 1.

The pasting properties of each GF flour blend formulation, were tested using a micro-doughLAB 2800 (Perten Instruments, Sidney, Australia), following the manufacturer’s cooking protocol with some modifications [43]. The mixing rate was constant and performed at 63 rpm during 43 min (2580 s). Additionally, the temperature cycle was: 30 °C for 6 min, followed by temperature rising until 90 °C during 15 min, with 90 °C constant for 7 min, then decreasing to 50 °C during 10 min and finally, being kept at a constant temperature of 50 °C for more 5 min, until the test ended. 4.00 g ± 0.01 of each flour blend sample (with HPMC) was weighed and placed in the chamber of micro-doughLAB. The amount of water used in the test was studied previously (method described above), to find the optimum water absorption. Tests were performed at least in triplicate.

#### 2.4.2. GF Dough Rheology

Dough samples from the three developed formulations (C, A17 and A23) were characterized through dynamic rheometric measurements—Small Amplitude Oscillatory Shear Measurements (SAOS), using a controlled stress rheometer (MARS III, Haake, Germany) coupled with a UTC–Peltier system, to perform rheology tests. It used a serrated parallel-plate PP20 (diameter of 20 mm), where the gap between plates was 1.0 mm (previously optimized for bread dough).

Each dough sample was prepared as presented in Section 2.3 (Table 1), in a small portion of dough, and after 50 min in a fermentation chamber Arianna XLT133 (Unox, Cadoneghe, Italy) at 29 °C, dough was permitted to rest at 5 °C during 30 min for stabilizing and to stop fermentation. Fermented dough sample was placed between the rheometer plates, and edges were covered with liquid paraffin to avoid sample drying throughout the running of the test.

Stress sweep tests at a frequency of 6.28 rad s^−1^ were accomplished to identify dynamic linear viscoelastic region of each sample. Then, frequency sweeps (from 0.00628 rad s^−1^ to 6.28 rad s^−1^) were performed at constant shear stress of 10 Pa. Temperature was maintained at 5 °C during the measurement to prevent dough fermentation. Before starting the test, samples were left for 15 min, to stabilize the temperature, and allow relaxing of residual stress. All rheology measurements were run at least in triplicate.

### 2.5. GF Bread Technological Performance

Bake Loss: was determined using the following expression [44]:Bake loss (%) = (W_bb_ − W_ab_/W_bb_) × 100(1)
where W_bb_ is the weight of the dough before baking and W_ab_ is the bread weight after baking and cooling.

Specific Bread Volume: was determined using the rapeseed displacement method [45].

Water activity (a_w_): was measured with an HygroPalm-HP23 (Rotronic Measurements Solutions, Switzerland), after mincing interior crumb from bread slices into the sample holder.

Bread Crumb Texture Profile Analysis (TPA): was performed after baking (2 and 24 h), in a texturometer TA.XT.plus (Stable Micro Systems, Surrey, UK). Bread slices with a thickness of 20 mm were prepared to perform a “two bite test” (double puncture test). Test conditions were: 25 mm diameter acrylic cylindrical probe, 1 mm s^−1^ of crosshead speed, 8 mm of penetration distance, and 5 s of waiting time. As previously described by other authors, for bread characterization, the degree of firmness and cohesiveness were considered the main representative texture parameters obtained from TPA [35].

Bread Crumb Color: was measured using a colorimeter Croma-Meter CR 400 (Konica-Minolta Sensing Americas, New Jersey, USA), by CIE (International Commission on Illumination) L*a*b* system with following parameters: L*—lightness (L* = 100 white, L* = 0 black), a*—intensity of green (−60 < a* < 0) or red (0 < a* < +60), b*—intensity of blue (−60 < b* < 0) or yellow (0 < b* < +60). In order to measure the change in the visual perception of the bread’s color, Delta E (ΔE*) was calculated using the expression:ΔE* = (ΔL*^2^ + Δa*^2^ + Δb*^2^)^1/2^(2)

### 2.6. GF Bread Nutritional Composition

Breads’ nutritional composition was evaluated in triplicate, after drying and grinding bread samples (all the results are presented in g 100 g^−1^ DW). Samples were analyzed for ash, protein, crude fat content, and fiber according to the Association of Official Analytical Chemists methods [46]. Breads were dried in an convection oven (WTC binder, Tuttlingen, Germany) for 24 h. The remaining moisture was, respectively (in g 100g^−1^): control 3.08 ± 0.10; A17: 3.07 ± 0.22; A23: 3.11 ± 0.10. Samples were stored in food safe plastic bags (with 50 g), at room temperature and protected from light.

Bread Moisture Content: was determined using an automatic moisture analyzer PMB 202 (Adam Equipment, Oxford, MS, USA), through gravimetric measurements, when reaching constant weight at 130 °C.

Total Ash Content: was measured gravimetrically, incinerating bread samples at 550 °C in a muffle furnace L 9/R (Nabertherm, Lilienthal, Germany).

Crude Fat Content: was determined by using petroleum ether in a Soxhlet extractor Det-gras N (JP Selecta, Barcelona, Spain), dried in the oven (overnight), and finally fat content was determined gravimetrically.

Total Protein Content: was determined using the Kjeldhal method. The nitrogen conversion factor used was 5.65, FAO’s reference for sweet potato, since it was the common ingredient used in the same share for the three bread samples.

Dietary Fiber: soluble, insoluble and total fiber of GF breads (C, A17 and A23) were determined by using a “Total Dietary Fiber Assay Kit” from Megazyme (Wicklow, Ireland), following manufacturer’s instructions. All procedures were performed at least in duplicate.

Carbohydrates: were determined by difference (100—% moisture—% ash—% protein—% fat—% fiber).

### 2.7. Total Starch and Resistant Starch Content

To calculate total starch content, an enzymatic procedure was used, as earlier presented by Goñi et al. [47], with some adaptations. Firstly, 100 mg of bread were weighed, adding 2 mL of KOH 2M, and shaking for 30 min at 4 °C. Next, 3 mL of Tris-maleate buffer 0.1 M (pH 6.9) was added, together with 1 mL of α-amylase and incubated for 45 min at 37 °C with constant shaking. Then, 3.0 mL of sodium acetate buffer 0.1M (pH 4.75) and 60 μL of amyloglucosidase were added, followed by incubation for 45 min at 60 °C with constant shaking. Next, 1.0 mL of this suspension, was centrifuged at 10,000 rpm for 15 min and 50 μL of the supernatant was collected and diluted with 950 μL MQ water and vortexed. Finally, starch was determined as glucose by DNS color reagent, with 540 nm reading in spectrophotometer, and converted to starch, multiplying by a conversion factor of 0.9.

Resistant starch content was assessed using the method proposed by Goñi et al. [47], slightly adapted, which started from weighing 100 mg of bread sample, centrifuging and adding 10 mL of HCl-KCl 0.1M buffer (pH 1.5). Afterwards, in order to hydrolyze protein, 200 μL of pepsin were added, followed by incubation at 40 °C for 60 min. Then, 9 mL of Tris-maleate buffer 0.1 M (pH 6.9) and 1.0 mL of α-amylase were added, with incubation at 37 °C for 16 h, with constant shaking. After centrifugation for 10 min at 10,000 rpm, the residue was allowed to separate, and the supernatant was discharged. The residue, containing the resistant starch, was washed and treated with 3.0 mL of KOH 2M to solubilize the remaining starch. Lastly, the solution was prepared with 5.0 mL of HCl 2M and 3.0 mL of sodium acetate buffer 0.4 M (pH 4.75), and incubated with 80 μL of amyloglucosidase to convert the starch into glucose. The starch calculation was performed as previously described.

### 2.8. In Vitro Starch Hydrolysis and Glycemic Index Estimation

In vitro starch hydrolysis and glycemic index estimation were determined according to Goñi et al. [47] and Germaine et al. [48]. Cooked gluten-free bread crumb portions corresponding to 1 g of starch, were chewed for 15 s, followed by rinsing the mouth with MQ water for 30 s, then collected to an Erlenmeyer. 1mL of pancreatic α-amylase (110 U mL^−1^ and sodium phosphate buffer pH 6.9) was added, and the mixture was kept in incubation, at 37 °C with constant shaking (120 rpm). Along the time of incubation, at 15, 30, 60, 90, 120, 180 min, aliquots of 2 mL were taken from the Erlenmeyer, boiled for 5 min to inactivate the enzyme and kept in an ice bath (4 °C). The DNS reagent was added to the supernatants of the aliquots and glucose content was determined using maltose standard curve at 540nm.

For eGI estimation, starch in vitro digestibility was graphically drawn, and area under the curve (AUC) was calculated using the trapezoids method. HI (starch hydrolysis index) of standard wheat bread = 100 and for the HI of each sample, HI = (AUC of sample bread/AUC of standard wheat bread) *100. eGI= 0.549×HI + 39.71 [47].

### 2.9. Antioxidant Capacity and Bioactive Compounds

Bioactive compounds (total phenols, *ortho*-diphenols, flavonoids contents) were analyzed using spectrophotometric methodologies previously reported in [49]. Radical scavenging activity was assessed using two methodologies: ABTS^•+^ and DPPH^•^, as well as ferric reducing antioxidant power (FRAP) assay, performed as described by [50] with minor modifications.

All analyses were completed in 96-well micro plates (Nunc, Roskilde, Denmark), using a microplate reader Infinite M200 (Tecan, Grödig, Austria), and were evaluated in triplicate (*n* = 3) for each sample.

#### 2.9.1. Extract Preparation

To perform bioactive compounds extraction from bread samples, 40 mg of each bread type previously dried and ground in powder were weighed and 1.5 mL of extracting solvent (methanol/distilled water (70:30, *v*/*v*) was added. Samples were stirred at the highest speed, for 30 min, at room temperature, and then centrifuged at 5000 rpm for 15 min at 4 °C. Supernatant was collected. The final volume was made up to 5 mL with methanol/distilled water (70:30, *v*/*v*). The procedure was repeated three times.

#### 2.9.2. Total Phenols Content (TPC) Determination

To assess total phenols content in bread samples, Folin–Ciocalteau reagent was used, with gallic acid as standard. This method consists in the reduction of phosphowolframate–phosphomolybdate complex through bread’s phenolic compounds, yielding blue reaction products. Briefly, 20 μL of sample and 100 μL of previously diluted Folin–Ciocalteau reagent were mixed and vortexed. Then, 80 μL of sodium carbonate was added and mixed in a vortex. Finally, after 30 min of incubation in the oven at 40–45 °C in the dark, the absorbance was read at 750 nm. Results were expressed in milligrams of gallic acid per gram of dry weight (mg GA g^−1^ DW) [51].

#### 2.9.3. Flavonoids Content (FlC) Determination

Flavonoids in bread samples were assessed through the following experimental procedure: 24 μL of sample were mixed with 28 μL of sodium nitrite (50 g L^−1^). After waiting 5 min, 28 μL of aluminum chloride (100 g L^−1^) was added and the mixture reacted for 6 min. Next, 120 μL of sodium hydroxide (1 M), was added to the mixture, the microplate was shaken for 30 s, and absorbance was measured at 510 nm. Flavonoid content was quantified using catechin as standard, and results are presented as milligrams of catechin per gram of dry weight (mg CAT g^−1^ DW) [51].

#### 2.9.4. Ortho-Diphenols Content (ODC) Determination

*Ortho*-diphenols in bread were measured using the protocol previously reported by Gouvinhas el al. [51]. 40 μL of sodium molybdate were added to 160 μL of samples properly diluted. Mixtures were vortexed and protected from light during 15 min at room temperature, before reading the absorbance at 375 nm. *Ortho*-diphenols content was quantified using gallic acid as standard, and results were expressed as milligrams of gallic acid per gram of dry weight (mg GA g^−1^ DW).

#### 2.9.5. Determination of Antioxidant Capacity

DPPH antioxidant capacity was measured placing 10 μL of sample or Trolox standard and 190 μL of the DPPH to each well of the microplate. The mixture was incubated for 30 min in the dark at room temperature, and absorbance read at 520 nm. Inhibition of free radical DPPH^•^ was calculated using the equation:%inhibition = (Abs blank − Abs sample)/Abs blank × 100(3)

DPPH^•^ samples scavenging capacity was determined by interpolation of the calibration curve for Trolox, and results were expressed as mmol Trolox per g of dry weight (mmol Trolox g^−1^ DW) [50]. ABTS radical inhibition was assessed using 12 μL of sample or standard, and adding 188 μL of ABTS working solution. The microplate was allowed to rest at room temperature and protected from light, during 30 min, before reading absorbance at 734 nm. Inhibition of ABTS radicals was calculated using the previously mentioned Equation (3).

The antioxidant capacity of the extracts was determined by interpolation of the calibration curve for Trolox, and results were expressed as mmol Trolox per g of dry weight (mmol Trolox g^−1^ DW) [50]. To determine ferric reducing antioxidant power (FRAP), 20 μL of sample were placed in each well of the microplate followed by 280 μL of FRAP working solution. After incubating the reaction at 37 °C for 30 min with protection from light, the absorbance was read at 593 nm. Trolox was used as standard and results were expressed as mmol Trolox per g of dry weight (mmol Trolox g^−1^ DW) [50].

### 2.10. Sensory Evaluation

Sensory evaluation of bread samples was carried out in a tasting room, using a structured scale, with a group of randomly selected 40 non-trained panelists, from Instituto Superior de Agronomia Food Science Department (both students and staff), according to ISO 11136:2014 [52]. Panelists were asked to taste the bread samples and assess the following parameters: color, appearance, aroma, texture, taste, global appreciation and buying intention in a 5-point hedonic scale from like extremely to dislike extremely.

Ethical Statement: All subjects gave their informed consent for inclusion before they participated in the study. The study was conducted in accordance with the Declaration of Helsinki, and the protocol was approved by the Ethics Committee of LEAF Research Centre.

### 2.11. Statistical Analysis

Results were analyzed with OriginPro 2019 (OriginLab Corporation, MA, USA). The statistical tool chosen to compare experimental data, was analysis of variance (one-way ANOVA). In order to evaluate means differences comparison, Tukey Test was used in a confidence level of 95%, with differences between mean values, being considered statistically significant at (*p* < 0.05). All data were presented as mean value and respective standard deviation.

## 3. Results and Discussion

### 3.1. Apple Flour Characterization

Apple flour (AF) was analyzed in order to obtain its nutritional profile as well as bioactive compounds and antioxidant capacity. Results are presented in Table 2. According to Feliciano et al. [53] who studied nine different apple varieties, fresh apples had about 80% water. Apples were dried to obtain the flour, so the amount of each component increased proportionally with the effect of concentration, due to water reduction. As expected, apple flour composition revealed a high content in carbohydrates, about 76%. Apple’s carbohydrate compounds are mainly sugars, with fructose being the most abundant, followed by glucose, sucrose, arabinose, galactose and xylose [53,54]. Ash content was similar to the values obtained by Espinosa-Solis et al. [23], but it was quite high when compared to apple pomace flours in general [17]. As regards protein, AF revealed having around 1.34%, which was higher than many studies about apple, where samples were prepared using just the edible part (pulp without skin and seeds); in contrast, the AF used in this work was prepared with the whole apple (seeds and skin) included. This fact could explain the higher protein content, because seeds are identified as an important source of protein [31]. In terms of fat, as expected, the obtained value was low, and similar to data from USDA (2017–2018) [55]. Finally, concerning fiber, values were high, about 13%, which is in agreement with other studies, both in apple flour and the equivalent amount in fresh apple [23,53]. In order to easily compare the nutritional composition of BWF with AF, its nutritional profile (provided by the manufacturer) is also presented (in g 100g^−1^): 13.3 protein; 3.4 lipids; 61.6 carbohydrates; 10.0 total fiber.

It is generally agreed by different authors that apple is rich in bioactive compounds. The major groups found were: hydroxycinnamic acid, flavan-3-ols, procyanidin, anthocyanin, flavanols, and dihydrochalcone classes. The main compounds included: chlorogenic acid, quercetin glycosides, procyanidin B1 and B2, catechin, epicatechin, and phloridzin [53,56]. In particular, Rabetafika et al. [31] studied apple seeds independently, finding chlorogenic and caffeic acids among others, procyanidin, phloretin and phloridzin [31]. Moreover, according to Rabetafika et al. [31], apple peel revealed the similar compounds to apple pulp, although in much higher concentrations: chlorogenic acid, epicatechin, catechin, phloretin, phloridzin, quercetin glycosides, and cyanidin. Therefore, from Table 2 it is possible to conclude that total phenolic, *ortho*-diphenols and flavonoids content were really high. Comparing with other authors, TPC values obtained for apple flour were within the range of values obtained for the peel (7.08-12.33 in mg GAE g DM^−1^) from different apple varieties, and much higher than the values presented for the apple cortex from the same varieties [57]. This shows the importance of the bioactive compounds in apple peel, as mentioned previously, especially given their potential health benefits [30,32]. Feliciano et al. [53] and Serra et al. [29], obtained lower values for TPC in fresh apple from the same variety, with values (in mg GAE g DM^−1^), respectively, of 7.61, and 6.80. Moreover, in a research study that studied seedless apple flour, TPC was even lower, 5.48 GAE g DM^−1^ [23], highlighting bioactive compounds in seeds increase apple flour TPC content, when compared with samples without seeds [31]. Regarding antioxidant capacity, ABTS methodology revealed a higher value comparing with DPPH, respectively 0.108 ± 0.011 and 0.073 ± 0.007 mmol Trolox g^−1^ DW, which difference could indicate a higher content in hydrophilic antioxidants, since DPPH does not identify these type of antioxidants due to the fact that is not soluble in water [58]. The antioxidant capacity assessed by ABTS of the previously mentioned seedless apple flour, was 0.004 mmol Trolox g^−1^ DW, which is a much lower value that our results. Regardless of differences in methodologies, the absence of seeds could be one of the reasons for the difference in these values [23]. Regarding FRAP, the obtained values were within the scope presented by Sethi et al. [57] for apple peel, which ranged from 0.05 to 0.15 mmol Trolox g^−1^ DW.

### 3.2. Gluten Free Dough Properties

#### 3.2.1. Mixing and Pasting Curves

After preliminary experiments, a peak torque value of 48 mN·m ± 4%, corresponding to 95.0% WA, was considered as optimum for better baking performance of the control formulation. A17 and A23 formulations were kept in this range of torque (±4%) by modifying the water addition. From micro-doughLAB mixing curves, water absorption value (14% moisture basis) of A17, and A23 formulations was, respectively, 81.0%, 79.0%.

In Table 3 it is possible to observe the results obtained from mixing curves and also torque values from pasting curve results. Dough development time (DDT) represents the mixture time needed for flours, together with water, to form the dough, and corresponds to the peak (maximum) torque. Stability is the time that dough holds onto its maximum peak torque, and softening is the decrease of torque after maximum value. According to Cornejo and Rosell [59], the interaction of sugar with starch changes the structure, leading to more compacted gels. Stability was also positively influenced by the introduction of apple flour, as well as its incorporation level. Thus, A17 and A23 presented an increase of stability values in comparison with C (control without AF). On the other hand, the softening has shown a higher value for A17 and A23, which means a greater decrease after the maximum torque, showing this negative impact of apple flour addition in the dough.

Regarding pasting curves, C1 represents the maximum torque of mixing flours, C2 is related to protein weakness, C3 is starch gelatinization, C4 concerns gel stability and amylase activity, and C5 corresponds to starch retrogradation [43]. When preforming pasting, the maximum torque of the flour blends’ mixing curves (C1) is obtained with a constant temperature of 30 °C. After 6 min, the temperature starts rising, reaching C2, which corresponds to protein weakening. This occurred at 70 °C, and the three flour blends had a similar viscosity response, around 10 m·Nm and at the same time. Following the heating process, viscosity increased until reaching the peak (C3) due to starch granules water retention and swelling, which led to starch gelatinization.

Gelatinization viscosity (C3) reached 52, 60, and 55 m·Nm of torque, respectively, for the C, A17 and A23 flour blends. The corresponding temperature was 83 °C for C and A17, while for A23 it was 90 °C. From these results it was possible to understand that AF contributed to increase gelatinization torque value, and also the temperature. Bread is a very complex matrix, with interaction among all the ingredients, which led to different dough behaviors. Particularly, sugars revealed an effect of concentration; however, the obtained values could not exclusively be attributed to sugars’ role in the bread, but also to the interactions between proteins, starch, HPMC and fibers [14]. Rocha Parra et al., investigated the pasting properties of starch-apple pomace mixtures, and reported that the reduction of starch as consequence of apple pomace lower replacement levels were compensated by the fiber content, which could explain the similarity in paste behavior of A17 and A23 to C [60]. Moreover, dietary fibers have a high water holding capacity, which could have led to the decrease of free water content that is responsible for facilitating the movement of particles, leading to the viscosity increase [18].

Next, starch granules reach a rupture point due to the maximum water absorption with temperature at 90 °C, and viscosity started decreasing until a minimum (C4). The torque values were 48 m·Nm for C flours blend and 36 m·Nm for both A17 and A23. This viscosity decay is the breakdown (difference between C3 and C4), which shows the resistance to heat and shear of the dough, and it is lower with higher heat resistance. This was seen in C, where breakdown viscosity was really low, and increased with the addition of AF, with values, respectively, of 24.0 and 19.0 for A17 and A23. This could be explained due to replacing of BWF which has more resistance to heat and shear stress when compared with apple flour, ascribed to its higher starch and protein content [61].

Finally, when temperature started decreasing, viscosity kept increasing and reached a maximum with starch retrogradation in the end of the test (C5). Retrogradation torque revealed a progressive decrease from C to A17 and to A23, respectively, 78, 71, 65 m·Nm, attributed to the reduction of protein and starch content in the flour blends, according to the different levels of AF incorporation in the dough. Since retrogradation is the reorganization of molecules and restructuring of amylose, the flours blend with apple, leading to lower values due to the less structured matrix. As previously mentioned, the interaction between all bread ingredients influences the GF dough rheology, affecting also starch pasting properties [7]. Setback viscosity is the difference between C5 and C4, which was respectively 30.0, 35.0, 29.0 m·Nm of torque, for C, A17, and A23, and is an indicator of the retrogradation starch capacity.

#### 3.2.2. Gluten Free Dough Rheology (SAOS)

In Figure 1A, the mechanical spectra of GF doughs (C, A17 and A23) are shown, representing the variation from elastic modulus (G’) and viscous modulus (G’’) as function of angular frequency (ω). Since all mechanical spectra have a similar shape, with the objective of better understanding the comparison between samples, the G’ values obtained at 6.28 rad s^−1^ (1 Hz) are presented in Figure 1B. For the three GF doughs, G’ was higher than G’’, which revealed a prevalence of the elastic behavior, along all frequency ranges. It is possible to observe that G’ and G’’ are frequency dependent, and at the same time, AF supplementation influenced the viscoelastic behaviour of the doughs A17 and A23. When adding AF to GF dough, G’ revealed a higher growth, when compared to G’’, showing a greater influence in the storage modulus (G’) and lower in loss modulus (G’’). The increase in G’ was progressively higher in A17 and A23, when compared with C. This could be ascribed to the replacement of BWF by AF, owing to the composition of each flour, respectively replacing protein and starch by sugars and fibers, which reinforces the structure. Comparing these results with Rocha Parra et al. [16], who used apple pomace levels of incorporation similar to the present study (17.8% and 20%), they also obtained a higher G’ than G’’, in a formulation with rice, cassava and egg white. According to the same authors [16], the higher concentration in polymers and fructose of AF is responsible for this behavior. Moreover, comparing this with specific volume and texture results, it is possible to understand that AF reduces air incorporation capacity, leading to a more compacted structure, and thus an increase of the viscoelastic functions.

### 3.3. GF Bread Technological Characterization

In Table 4, the results obtained from GFB analysis are presented. Baking loss is the weight difference between the dough before baking and the baked bread, representing the water evaporation during baking. In the three studied breads (C, A17, A23), there were no significant (*p* > 0.05) differences among them. Similar results were obtained in previous work, where acorn flour replaced similar percentages of buckwheat flour (23% and 35%), and differences were not found between the tested breads [36].

Regarding bread’s specific volume, it was possible to perceive that the values significantly (*p* < 0.05) decreased from control bread to both AF incorporation levels. These results shown that the incorporation of AF developed a more consistent bread, having a lower volume per weight unit. This could be ascribed to the fiber increase, besides starch and protein content reduction, owing to the replacement of buckwheat by apple flour, which reduces air incorporation ability, leading to a bread’s lower specific volume. These results are in accordance with other authors who have studied apple pomace incorporation in GF bakery, which particularly indicate a reverse relation between volume and firmness, as we also found in the present work [18,20,21].

Analyzing the results obtained for texture parameters, it was possible to observe that firmness value at 2 h increased from C to A17, and decreased when raising the level of apple flour incorporation to A23, being all different (*p* < 0.05). Regarding firmness at 24 h, the values for the three different breads were also different (*p* < 0.05), showing a similar behavior as previously described in firmness at 2 h. These results were aligned with the reduction of the volume previously explained, attributed to the effect that AF generates in the bread. The first level of the tested flour (17%) increased bread’s firmness and accordingly reduced the specific volume when compared to control bread. The higher amount of incorporation (23%) revealed also a firmness increase, comparing with control, but with lower values than A17. It seems that the level of added flour reveals higher impact in the firmness increase than in specific volume decrease, which could be explained due to the interaction between fiber increase, starch and protein reduction, and consequent reduction in air incorporation ability. This type of firmness performance has also been identified with other apple-sourced ingredients tested in GF formulations [18,21]. Cohesiveness at 2 h was not significantly different between the control and the apple flour incorporations. Cohesiveness at 24 h showed a reduction from C to A17, being not significantly different, and from A23, but control and A23 were significantly different (*p* < 0.05). Water activity significantly decreased (*p* < 0.05) with the incorporation of apple flour, among the three breads C, A17, and A23. AF has about three times less moisture when compared with BWF, so the replacement of 17% and 23% by AF explains the water activity reduction, since the water available is lower in apple flour-containing breads. Additionally, the water absorption was also higher in C bread (95%) than in A17 (81%) and A23 (79%), which together with the starch, protein, sugars and fiber content of each formulation, also explains the differences in water activity, due to the water movements and availability along the bread’s microstructure [22].

Regarding bread crumb color, the L* parameter reduced with the incorporation of AF, with difference (*p* < 0.05) between the control and both A17 and A23. These results showed a reduction of lightness in bread, meaning a slightly darker color, as we can observe from Figure 2. This fact could be explained by caramelization and Maillard reactions, occurring during the baking process, since AF bread presents higher content in reducing sugars [62]. Following the opposite trend, a* and b* parameters were significantly increased (*p* < 0.05) by incorporation of AF, but without differences between AF breads, revealing more reddish and yellow color. Similar color parameters were also obtained in other studies where apple-derived ingredients were added to bakery products, with bread crumb darkness increasing as the main effect [16,18,21].

According to Rogowska [63], ΔE* values mean the following perception to human eyes: ΔE* ≤ 2.0, not perceptible; 2 < ΔE* < 3.5, an inexpert observer could see a change between colors; 3.5 < ΔE* < 5, anyone could see some difference between colors; ΔE* > 5, everyone can recognize two different colors. The obtained results for the bread crumb colors comparison (ΔE*) were: ΔE*(control—A17%) = 2.38; ΔE*(control—A23%) = 2.43; ΔE *(A17%—A23%) = 0.19. Consequently, it is possible to understand from the ΔE* values, and observing Figure 1, that apple flour introduced a change in control bread’s color. Nevertheless, as between A17 and A23, the color did not have perceptible visual changes.

### 3.4. Nutritional Bread Analysis

Regarding GFB nutritional analysis, the results are presented in Table 5.

As regards moisture, the results have shown a significant (*p* < 0.05) decrease from control to A17 and A23. As mentioned in Section 3.3, BWF presents higher moisture content, 10.6%, when compared to AF with 3.5%. Moreover, C bread (95%) revealed an increased water absorption in comparison to A17 (81%) and A23 (79%), contributing as well to higher bread moisture.

From the values in Table 5, it is possible to observe that ash content was high in all breads in study. Thus, when comparing with other authors using apple products, their studies shown much lower ash values than in the present study [16]. These high values were attributed to BWF and AF richness in minerals.

The protein content was different (*p* < 0.05) between control bread and sweet AF breads, decreasing from C to A23, but not showing significant differences between A17 and A23. According to Hager et al. [61], which is in accordance with manufacturer information, BWF has between 12% and 13% of protein content. On the other hand, from our results, AF presented 1.34 g 100 g^−1^. Accordingly, when replacing 17% and 23% of BWF by AF, bread’s protein level was expected to undergo a reduction. Comparing our results with other studies about the addition of apple pomace in rice flour GF cakes, and apple pomace in brown rice crackers, a reduction of protein content was also observed [15,18].

As concerns fat content, the results showed differences between the breads. In particular, C and A23 breads differ in their fat content (*p* < 0.05) whereas no differences were observed between C and A17, and between breads with different levels of apple flour incorporation. This could be due to the increased amount of sugar from apple flour, which triggered glucose catabolite repression, and under such conditions, yeast cells direct used acetyl-CoA for lipid synthesis rather than the TCA (tricarboxylic acid) cycle [64,65]. In general, GFB presents a higher fat content, which is not nutritionally recommendable, but in the present research, the bread content of fat was lower than the average found in other studies [66].

Total fiber in breads A17 and A23 increased when compared with the control (C), as was expected, since AF has a higher content in total fiber, around 12.85%, while BWF presents around 2.20%. Thus, the replacement of BWF by AF led to a sweet GFB with higher fiber content. Insoluble fiber was higher in A17 and A23 also due to differences between BWF and AF insoluble fiber content. Other authors studying apple-derived ingredients incorporation in baking products, also obtained an increase in fiber content [18,24]. The amount of fiber in GFB is an important aspect, both for its well-known health benefits, and also its particular influence in GF rheology dough behavior, as mentioned in Section 2.2. Several authors have concluded that GF bakery products in general, are low in fiber content, an aspect that is recommended to be improved due to particular celiac consumers’ health condition [7,8,67]. Moreover, fiber content plays an important role in water movements and interaction between components, which will influence GF dough behavior, and consequently bread characteristics [22]. Concerning soluble fiber, there were no significant differences between C, A17, and A23. The three developed breads have a higher content in fiber when compared with the results of different studies with GFB [7,66,67].

Regarding carbohydrates, the amount is quite similar between the three breads C, A17 and A23, but with different profiles. According to the results in the next section, as shown in Table 6, it is evident that control bread has higher starch content, while A17 and A23 have lower starch and higher fructose content (from apple flour addition), which is also in accordance with the next section’s eGI results.

### 3.5. Total and Resistant Starch Content, In Vitro Starch Hydrolysis and Glycemic Index Estimation

In Table 6 the levels of total, resistant and digestible starch as well as hydrolysis index (HI), and estimated glycemic index (eGI) are represented. It is possible to observe that total starch significantly decreased (*p* < 0.05) when BWF was replaced by AF (A17 and A23), since the latter ingredient has a low starch content. For the same reason, the resistant starch value was also reduced significantly (*p* < 0.05) in breads A17 and A23, when compared with C.

The postprandial response is related to carbohydrate digestion, and can be estimated by the glycemic index (eGI), through a model behavior developed to allow comparison of starchy foods [47,48]. Glycemic index can be defined as the increasing area under the curve (AUC) of the blood glucose concentration, occurring as a response to carbohydrate-containing food ingestion, in relation to white bread with GI = 100 as a food reference [68]. Foods can be categorized by its eGI value: low eGI (<55), intermediate eGI (55–70) and high eGI (>70) [69]. eGI can be estimated through mimicking through an in vitro digestion process, using the AUC ratio between the food under study and food reference.

Table 6 show that HI significantly increased (*p* < 0.05) when breads were supplemented with AP, and consequently, eGI also increased. This result was expected since sweet GFB (A17 and A23) presents higher fructose and total carbohydrates content (Table 5), in comparison with C bread. In general, GFB is characterized by high eGI and low RS [7]. Additionally, in a study with twelve GF breads from Brazilian market, the value of eGI ranged between 61 and 80 [67]. Nevertheless, the developed sweet breads A17 and A23 revealed a medium eGI, close to the low category, respectively 56.15 ± 0.36; 56.81 ± 0.79, which was one of the objectives of the present work. Moreover, it is important to mention that bread characteristics, as endogenous factors and macroscopic structure of the food matrix, type of starch, protein and lipid content, influence glycemic response [66]. Furthermore, polyphenols and flavonoids could act as inhibitors of alfa-glucosidases, contributing to lower starch hydrolysis and consequently eGI. Thus, the eGI of sweet breads, A17 and A23 were medium, just 1 point ahead of low eGI, attributed to the high fraction of RS, bioactive compounds and fiber content. Additionally, research about GFB reveals that increasing bread fiber is a balanced strategy to reduce eGI, instead of higher levels of fat, as seen quite often [7,66]. According to WHO [70], decreased food eGI is essential to reduce the risk of developing diabetes, obesity, cardiovascular diseases and some types of cancer; this is particularly important in bread since it is a staple food. Gorjanović et al. [32] concluded, for example, that apple pomace decreased the glycemic index, reducing the risk of obesity and diabetes.

### 3.6. Bioactive Compounds and Antioxidant Capacity

The developed GFB bioactive compounds’ content and antioxidant capacity are presented in Table 7. The levels of all analyzed bioactive compounds increased when apple flour was added, in comparison to control, being significantly different (*p* < 0.05) for both levels of apple flour incorporation, and for all compounds group analyzed: total phenols, *ortho*-diphenols and flavonoids content. As regards antioxidant capacity, it can be seen that for both apple flour levels, and for the three determination methodologies, the values were significantly (*p* < 0.05) higher than for control bread. For DPPH, ABTS, and FRAP determinations, no significant differences between A17 and A23 were found (*p* > 0.05). According to different authors, apple is a source of bioactive compounds and antioxidants [53,56]. Thus, its incorporation in GFB, as expected, led to the raise of total phenols, *ortho*-diphenols and flavonoids content as well as antioxidant capacity. Other authors obtained also the increase of bioactive compounds and antioxidant capacity [15,17,20,23,24]. In particular, Mir et al. [15] obtained much lower values, respectively, 0.82 mg GA g^−1^ and 0.063 mg Catechin g^−1^ for total phenols and flavonoids content, for its higher level of incorporation in crackers, nevertheless it was 9%, much lower than in the present work. Espinosa-Solis et al. [23] obtained quite similar values for cooked pasta, replacing 50% of wheat by apple flour, with 0.025 mmol Trolox g^−1^ for ABTS antioxidant activity [23]. Concerning GFB characteristics and health conditions of celiac patients, it is important to increase bioactive compounds levels in bread, due to their potential benefits to health [71]. Nevertheless, it is important to highlight the influence of fermentation, mixing, baking time and temperature in antioxidant activity and also in bioavailability of bioactive compounds [14].

### 3.7. Sensory Evaluation

Results of sensory evaluation are represented in Figure 3. Concerning sensory analysis, in a hedonic scale from 0 (dislike) to 5 (like very much), texture score was similar for the three breads (4), with overall acceptance and flavor raised from 3 (in control bread) to 4 (in both A17 and A23). In terms of overall acceptance, panelists preferred breads with 17% and 23% of apple flour compared with the control bread, with a score of 3.8, 3.7 and 3.3, respectively. Also, flavor was preferred in A17 and A23 comparing to C, with respectively 3.7, 3.6, and 3.0 points for each sample. Nevertheless, the texture of the control bread had a higher score, which is aligned with the results obtained for dough rheology and bread texture and volume (A17 and A23 with lower volume and higher crumb firmness values). However, breads with higher AF incorporation level presented higher scores, and it would not have been only due to the sweet flavor of the GFB, but also owing to other AF characteristics, with positive influence in flavor and taste, which is also mentioned by other authors that have been studying the potential of apple-sourced ingredients. In all the other evaluated characteristics (color, appearance, aroma, and texture) there were lesser differences between the three breads. Buying intention was also analyzed; 12, 18 and 22 panelists stated a buying intention of 4 and 5, respectively to C, A17 and A23, corresponding to 30%, 45%, and 55%. Thus, the preference scores have shown the importance of overall acceptance and flavor in the moment of consumer’s choice. Comparing with other research with apple-derived ingredients addition, sensorial evaluation also revealed positive results [20,21].

## 4. Conclusions

Supplementation with apple flour presents promising results in developing bakery products with potential health benefits and innovative sensory attributes, especially in sweet breads, cakes and cookies. At the same time, with apple flour it is possible to follow the trend of rich-fiber, natural health promoting ingredients, sustainability and also deliver apple characteristics to the bread.

Apple flour improved technological and sensory characteristics of sweet gluten-free (GF) bread, being a source of bioactive compounds and antioxidants. The starch hydrolysis index increased, as well as eGI, nevertheless with values really close to the low eGI category (56). Nevertheless, the results have demonstrated apple flour’s impact on dough’s pasting properties and rheology (increasing G’ and G’’), leading to a more compacted dough structure and breads with higher firmness. Sensory results show that this impact on texture was compensated by the positive effect on flavor.

Furthermore, the apple flour samples presented a high concentration of bioactive compounds; nevertheless, from the sustainability perspective, it is important to study its bioavailability. Apple flour is a valuable by-product that can be used as natural sweetener in GF bakery products, giving a sweet gluten-free bread with better acceptance by the consumer, rich in fiber, antioxidants and other bioactive compounds. In conclusion, apple flour seems to be an ingredient with a positive impact in GF bread quality in terms of nutrition, bioactivity and sensory properties, while at the same time it is a resource saved from being wasted.

## Figures and Tables

**Figure 1 foods-11-03172-f001:**
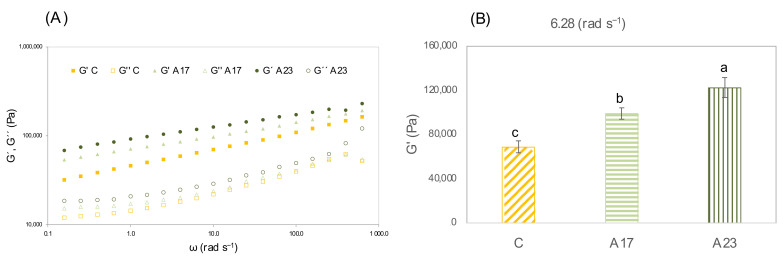
(**A**) Mechanical spectra of GF doughs (C, A17, A23); G′ (storage modulus—filled symbol), G′′ (loss modulus—open symbol). (**B**) G′ values at 6.28 rad s^−1^ (1 Hz) for the same samples. Means with different letters are significantly different (one-way ANOVA, *p* < 0.05).

**Figure 2 foods-11-03172-f002:**
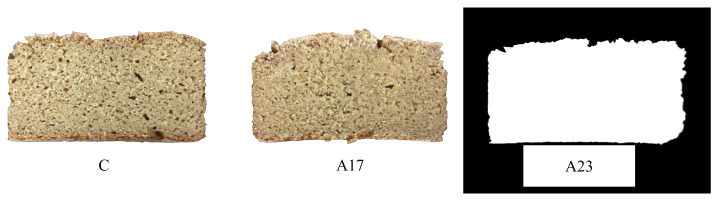
GF breads: (C) control, without apple flour, (A17) with 17%, and (A23) with 23% apple flour addition.

**Figure 3 foods-11-03172-f003:**
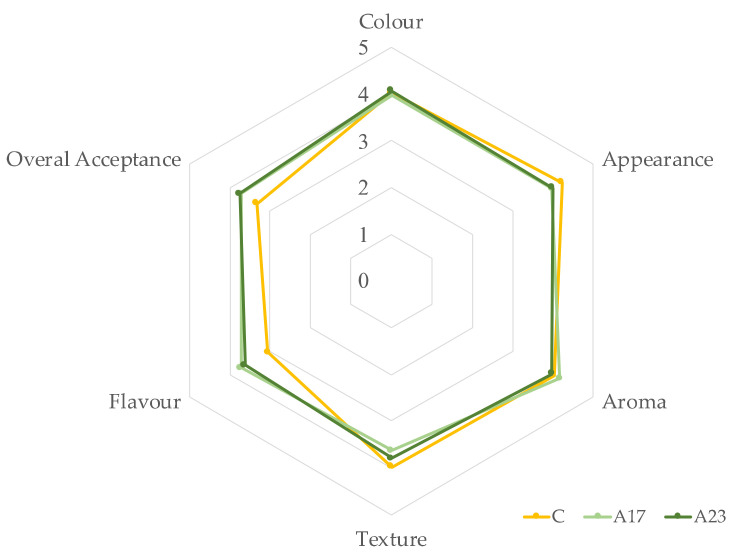
GF breads: Sensorial analysis of sweet GFB, respectively (C) control, without apple flour, (A17) with 17%, and (A23) with 23% of apple flour incorporation.

**Table 1 foods-11-03172-t001:** Formulation of the gluten-free dough (GFD) samples and respective codes.

Ingredients (%)	Control (C)	Apple 17% (A17)	Apple 23% (A23)
Buckwheat flour	46.0	29.0	23.0
Sweet potato flour	29.0	29.0	29.0
White lupin flour	25.0	25.0	25.0
Apple flour	0.0	17.0	23.0
Cinnamon (in relation to flours)	1.7	1.7	1.7
Sunflower oil (in relation to flours)	5.5	5.5	5.5
HPMC (in relation to flours)	4.6	4.6	4.6
Dried yeast (in relation to flours)	2.8	2.8	2.8
Sugar (in relation to flours)	2.8	2.8	2.8
Salt (in relation to flours)	1.5	1.5	1.5
Water absorption (14% moisture basis) *	95.0	81.0	79.0
Amount of water (in relation to flours after adjustment to 14% moisture basis)	107.5	95.5	93.7

* Water absorption determined by micro-doughLab mixing curves (Section 2.4.1).

**Table 2 foods-11-03172-t002:** Apple flour characterization: nutritional profile, bioactive compounds, and antioxidant capacity.

Apple Flour	**Moisture**	**Ash**	**Protein**	**Lipids**	**Total Fiber**	**Ins. Fiber**	**Sol. Fiber**
	**(g 100 g^−1^ DW)**						
	3.98 ± 0.42	5.47 ± 0.14	1.34 ± 0.11	0.49 ± 0.02	12.85 ± 0.21	10.00 ± 0.28	2.85 ± 0.07
	**Carbohydrates** **(g 100 g^−1^ DW)**	**TPC**	**ODC**	**FlC**	**ABTS**	**DPPH**	**FRAP**
		**(mg GA g^−1^ DW)**		**(mg Cat g^−1^ DW)**	**(mmol Trolox g^−1^ DW)**		
	76.17 ± 0.62	8.86 ± 0.50	6.83 ± 0.58	9.46 ± 1.02	0.108 ± 0.011	0.073 ± 0.007	0.069 ± 0.002

Ins. Fiber: insoluble fiber, Sol. Fiber: soluble fiber, TPC: total phenols content, ODC: *Ortho*-diphenols content, FlC: flavonoids content, DPPH: diphenyl-1-picrylhidrazyl radical, ABTS: 2,2-azino-bis (3-ethylbenzothiazoline-6-sulphonic acid) diammonium salt, FRAP: ferric reducing antioxidant power.

**Table 3 foods-11-03172-t003:** Torque values (mN·m) and respective temperatures (°C) along pasting curves for GF flour blends: (C) control, without apple flour, (A17) with 17%, and (A23) with 23% apple flour addition.

	C		A17		A23	
WA	95.0%		81.0%		79.0%	
DDT (min)	0.95		1.09		1.19	
Stability (min)	0.85		1.25		1.40	
Softening (mN·m)	9.0		12.5		12.5	
	Torque (mN·m)	T (°C)	Torque (mN·m)	T (°C)	Torque (mN·m)	T (°C)
	C		A17		A23	
C1	47.0	30.0	47.0	30.0	47.0	30.0
C2	9.0	70.0	11.0	67.0	10.0	75.0
C3 (gelatinization)	52.0	83.0	60.0	83.0	55.0	90.0
C4	48.0	90.0	36.0	90.0	36.0	90.0
Breakdown (C3–C4)	4.0	-	24.0	-	19.0	-
C5 (retrogradation)	78.0	50.0	71.0	50.0	65.0	50.0
Setback (C5–C4)	30.0	-	35.0	-	29.0	-

**Table 4 foods-11-03172-t004:** Physical characteristics of GF breads: (C) control, without apple flour, (A17) with 17%, and (A23) with 23% apple flour addition.

		C	A17	A23
Baking loss (%)		11.69 ± 0.80 ^a^	11.89 ± 0.81 ^a^	11.72 ± 0.79 ^a^
Specific Volume (cm^3^/g)		1.38 ± 0.11 ^a^	1.18 ± 0.07 ^b^	1.13 ± 0.07 ^b^
Firmness 2h (N)		21.01 ± 2.60 ^c^	26.34 ± 2.21 ^a^	23.88 ± 2.40 ^b^
Firmness 24h (N)		24.01 ± 2.70 ^c^	28.58 ± 1.70 ^a^	26.77 ± 3.33 ^b^
Cohesiveness 2h		0.51 ± 0.04 ^a^	0.46 ± 0.04 ^a^	0.45 ± 0.07 ^a^
Cohesiveness 24h		0.47 ± 0.02 ^a^	0.43 ± 0.04 ^a,b^	0.41 ± 0.05 ^b^
a_w_		0.957 ± 0.00 ^a^	0.95 ± 0.01 ^b^	0.93 ± 0.00 ^c^
Crumb Color	L *	53.69 ± 1.24 ^a^	51.83 ± 0.94 ^b^	51.90 ± 0.87 ^b^
	a *	5.87 ± 0.22 ^b^	6.94 ± 0.25 ^a^	7.10 ± 0.28 ^a^
	b *	27.53 ± 0.54 ^b^	28.55 ± 0.55 ^a^	28.63 ± 0.63 ^a^

^a, b, c^ Means with different letters on the same column are significantly different (one-way ANOVA, *p* < 0.05).

**Table 5 foods-11-03172-t005:** Proximate composition: moisture, ash, protein, fat, and fiber (in g 100 g^−1^ DW) of the GF breads: (C) control, without apple flour, (A17) with 17%, and (A23) with 23% apple flour addition.

	Moisture	Ash	Protein	Fat	Total Fiber	Insoluble Fiber	Soluble Fiber	Carbohydrates
C	49.71 ± 1.35 ^a^	3.88 ± 0.04 ^a^	14.60 ± 0.71 ^a^	4.25 ± 0.32 ^b^	18.30 ± 0.57 ^b^	16.70 ± 0.57 ^b^	1.60 ± 0.00 ^a^	56.52 ± 0.71
A17	47.18 ± 0.37 ^b^	3.78 ± 0.04 ^b^	12.19 ± 1.00 ^b^	5.86 ± 0.65 ^a,b^	19.63 ± 0.04 ^a,b^	18.05 ± 0.21 ^a,b^	1.58 ± 0.18 ^a^	56.21 ± 0.21
A23	45.93± 0.48 ^c^	3.72 ± 0.05 ^b^	11.44 ± 1.12 ^b^	5.94 ± 0.77 ^a^	19.95 ± 0.21 ^a^	18.85 ± 0.64 ^a^	1.10 ± 0.42 ^a^	56.17 ± 0.58

^a, b, c^ Means with different letters on the same column are significantly different (one-way ANOVA, *p* < 0.05).

**Table 6 foods-11-03172-t006:** Total Starch (TS), Resistant Starch (RS), Digestible Starch (TS-RS), Hydrolysis Index (HI) Estimated Glycemic Index (eGI) values of the GFB C, A17, A23.

	Total Starch (TS) (g 100g^−1^)	Resistant Starch RS (g 100g^−1^)	Digestible Starch TS-RS (g 100g^−1^)	Hydrolysis Index HI	Estimated Glycemic Index (eGI)
Control	33.41 ± 1.44 ^a^	14.80 ± 0.37 ^a^	18.67 ± 1.23	25.86 ± 0.24 ^b^	53.91 ± 1.23 ^b^
A17	31.48 ± 1.08 ^b^	11.65 ± 0.20 ^b^	19.83 ± 0.80	29.94 ± 0.66 ^a^	56.15 ± 0.36 ^a^
A23	31.15 ± 0.13 ^b^	10.60 ± 0.02 ^c^	20.56 ± 0.12	31.14 ± 1.44 ^a^	56.81 ± 0.79 ^a^

^a, b, c^ Means with different letters on the same column are significantly different (one-way ANOVA, *p* < 0.05).

**Table 7 foods-11-03172-t007:** Phenolic composition and antioxidant capacity of the GFB with the incorporation of different ratios of apple flour (A17% and A23%) and comparison with the control samples.

			C	A17	A23
Phenolic Composition	TPC	(mg GA g^−1^ DW)	3.166 ± 0.162 ^c^	3.842 ± 0.162 ^b^	4.241 ± 0.105 ^a^
	ODC		3.004 ± 0.102 ^c^	4.475 ± 0.113 ^b^	4.881 ± 0.116 ^a^
	FlC	(mg Catechin g^−1^)	0.962 ± 0.067 ^c^	1.390 ± 0.063 ^b^	1.434 ± 0.085 ^a^
Antioxidant activity	DPPH	(mmol Trolox g^−1^)	0.012 ± 0.001 ^b^	0.016 ± 0.001 ^a^	0.017 ± 0.001 ^a^
	ABTS		0.021 ± 0.001 ^b^	0.026 ± 0.001 ^a^	0.027 ± 0.001 ^a^
	FRAP		0.015 ± 0.001 ^b^	0.023 ± 0.002 ^a^	0.023 ± 0.001 ^a^

^a, b, c^ Means with different letters on the same row are significantly different (one-way ANOVA, *p* < 0.05). TPC: total phenols content, ODC: *Ortho*-diphenols content, FlC: flavonoids content, DPPH: diphenyl-1-picrylhidrazyl radical, ABTS: 2,2-azino-bis (3-ethylbenzothiazoline-6-sulphonic acid) diammonium salt, FRAP: ferric reducing antioxidant power.

## Data Availability

The data presented in this study are available on request from the corresponding author.

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
