# Peer review of "Apple Flour in a Sweet Gluten-Free Bread Formulation: Impact on Nutritional Value, Glycemic Index, Structure and Sensory Profile"

_foods, 2022, doi:10.3390/foods11203172_

Round 1

Reviewer 1 Report

In presented article studied the nutritional, technological, and sensory characteristics of sweet gluten-free bread with apple flour incorporation. The article is generally well written, however, some minor corrections are necessary:

Page 2, lines 76-79, add more novel references regarding apple powder addition to the bakery products,

Page 3, In the material and method section, it is unclear if the commercial apple powder was obtained as stated in line 109 or if the authors prepared apple powder as stated in lines 132-136.

What kind of apples was used?

What kind of drying processes was applied?

What was the moisture content of the apples and obtained four?

Page 5 and 8, mark formulas with numbers,

Page 13, calculate ΔE for color characteristics of gluten-free bread, add it to Table 4, and discuss.

Reviewer 2 Report

Lines 41-44: Please add reference for market research and forecasts.

Lines 53: Please rephrase. An addition of sugar for technological purposes that is consumed by yeast would not be expected to impact/increase GI but would possibly drastically improve technological bread quality

Lines 58-59: Text refers to WHO but reference mentions EFSA. Additionally, it might be better to reference the scientific opinion associated with the article in the news feed. These scientific opinions are published and can be cited with a DOI.

Lines 94-95: Please add a reference for the GI of SPF.

Lines 121-123 DPPH is mentioned twice.

Section 2.3: Please add information on batch sizes during mixing, amount of dough (batter) per loaf and ideally dimensions of the tins used.

Additionally, for certain measurements where the whole loaf is measured (e.g, bake loss and specific volume), a number of 3 breads per batch seems quite low. For measurements where slices have been evaluated, the number values obtained should be sufficient to characterise the batch adequately.

Lines 196-199: Please mention this optimised water level and whether it was the same level for all formulations (C, A17 and A23).

Lines 239-240: How were the breads dried and what was the remaining moisture content?

Lines 261-262: Please mention whether fresh or dried bread was used to perform the analysis (and how the bread was stored).

Lines 292-295: Was standard wheat bread measured also? Otherwise, how is the AUC of standard wheat bread obtained to include in the calculation? In general, please be careful to not refer to GI but only to eGI. These values do not represent real GI numbers and can only be compared amongst each other. (Please do not compare to actual GI values that are considered low, these are determined in a very different way.)

Line 376: It would be beneficial to the reader to make a statement about the expected carbohydrate profile amounting to 76% in total. What are the expected major carbohydrate compounds?

Table 2: If data are available, it would be beneficial to the reader to find compositional data of AF in comparison to those of BWF, since this is the flour being replaced in A17 and A23.

Lines 400-402: Please make it clearer which statements are referring to the literature and which statements refer to analysis performed for this article.

Lines 434-435: This publication is specifically mentioning wheat doughs and therefore referring to the water availability for gluten-network development. Please describe more in detail why water availability is still relevant here and how (which main ingredients). Or rather refer to a reference dealing with water availability in GF dough systems.

Lines 439-441: The degree of softening mainly matters when the mixing time goes beyond dough stability. How does the DDT determined in the experiments translate to the mixing conditions and times applied in the baking trials? The different DDTs in context with the constant mixing times applied should be discussed together with the bread quality characteristics.

Table 5: What are the different values referring to? The table caption mentions g/100 g DW as unit but the moisture content is also mentioned and the carbohydrate content calculated as 100-all other constituents? If the values for constituents are expressed in g/100g DW, how do they add up to 100 with the moisture content of the (presumably) fresh bread?

Table 6: Do values refer to dry or fresh samples? Total starch content here is in the range of 31 to 34 g/100 g. Please explain how this matches with the 10 to 15 g/100 g carbohydrate content presented in Table 5.

Author Response

Please the attachment.

Reviewer 3 Report

The work is good and has been well presented.

Title: indicates the work undertaken

Abstract: complete with clear aim, result and conclusion

Keywords: sufficient, except for circular economy this keyword might be changed 

Introduction: Concise with strong backdrop supporting the aim of work

Materials and Methods: Sufficient for replication

Results and discussion: All results have been well justified and supported with adequate references.

Conclusion: precise

Minor corrections have been highlighted in the manuscript which might be undertaken to enhance the quality of manuscript

Reviewer 4 Report

This study investigated the effects of apple flour on the nutritive value, glycemic index, structure, and sensory profile of the sweet gluten-free bread. The manuscript is well written, and the results are clear and interesting.

Comments:

1. In the title, please use “gluten-free” instead of “gluten free”.

2. Rewrite the abstract and add numerical data.

3. Some references were not cited in the manuscript, such as lines 41-44, 181, 358, 384, and 650.

4. For all tables, the letters should be placed after the standard deviations.

5. Line 537-538; “…, and decreased when raising the level of apple flour incorporation to A23, …”; Please explain the possible reason.

6. Line 793; Reference 22; The title of the article is wrong. 
